# Happy work: Improving enterprise human resource management by predicting workers' stress using deep learning

Yu Zhang 🔵 *, Ershi Qi

College of Management and Economics, Tianjin University, Tianjin, China

* yu_zhang_@tju.edu.cn

**Data Availability Statement:** All relevant data are within the paper and its Supporting information files.

**Funding:** The author(s) received no specific funding for this work.

## Abstract

Recently, workers in most enterprises suffer from excessive occupational stress in the workplace, which negatively affects workers' productivity, safety, and health. To deal with stress in workers, it is vital for the human resource management (HRM) department to manage stress effectively, bridging the gap between management and stressed employees. To manage stress effectively, the first step is to predict workers' stress and detect the factors causing stress among workers. Existing methods often rely on the stress assessment questionnaire, which may not be effective to predict workers' stress, due to 1) the difficulty of collecting the questionnaire data, and 2) the bias brought by workers' subjectivity when completing the questionnaires. In this paper, we aim to address this issue and accurately predict workers' stress status based on Deep Learning (DL) approach. We develop two stress prediction models (i.e., stress classification model and stress regression model) and correspondingly design two neural network architectures. We train these two stress prediction models based on workers' data (e.g., salary, working time, KPI). By conducting experiments over two real-world datasets: ESI and HAJP, we validate that our proposed deep learning-based approach can effectively predict workers' stress status with 71.2% accuracy in the classification model and 11.1 prediction loss in the regression model. By accurately predicting workers' stress status with our method, the HRM of enterprises can be improved.

## Introduction

With the rising of artificial intelligence (AI), the new technique has impacted various fields. Human resources management (HRM) in the high-technology enterprise, is being reshaped by developing the novel personal selection and management with neural network [1]. For a long time, managers in enterprises attempt to adopt the optimal method to recruit and manage the employees. Rather than the traditional recruit procedure consisting of cover letter, resume and interview, HR consultants commonly make the decision according to the social networking, online competitions and data exploration techniques, by which they are able to know the advantages and disadvantages of the candidates. Using neural networks is a new way to screen the accurate background and potentials of candidates, consequently fastening recruiters' work.

**Competing interests:** The authors have declared that no competing interests exist.

As the development of the neural network in HR, the neural network-based recruit is much more mature, comparing to the HR management for the management is usually affected by the difference of individuals.

From the view of enterprises manager, they commonly pay more attention to their working performance and the sense of belonging to enterprises. In the research of HRM, the scientific measurement to HRM policies include: involvement; training, development and education; work conditions; and compensation and rewards. Gisela Demo *et al.* [2] pointed out that if the appropriate psychometric parameters are involved, along with firm policies, the HRM would be more reliable and comprehensive. Additionally, it can be used to be a diagnosis to improve employees' well-being at work and optimize organizational results. From 2012, the research on psychometric parameters in HRM has attracted numerous attention from different aspects [3–5]. We notice that the previous works lack scientific analysis on the psychometric parameters, especially in the high-tech companies, that the majority of companies are the youth. Then, our work focuses on these employees' mental conditions as stress level. By using the neural network technique, our scheme is able to provide a stress level report for the manager to make a proper decision, which aims to release and eliminate the influence caused by the mental issues.

Different from the research on stress in the behavioral [6, 7] and health [8, 9] fields using social "theory" to explain the stress as an aspect of an individual's daily life, our work is aiming to describe the stress in the mathematic method by combining statistics and artificial intelligence technique. However, there is a challenge in the real-world applications, that is the continuous monitoring of an individual's state, involving physiological data collection, preprocessing, feature extraction coupled with selection/transformation to finally classify these stress-related features. The state-of-the-art of stress recognition algorithms such as the automatic stress recognition algorithms, the real-time non-intrusive fatigue monitor and a wearable emotional state assessing system, and a classification of emotions by neural network.

To optimize the accuracy of the model, it is required that we perform a comprehensive analysis of the machine learning paradigms available and choose the one with optimal predictive ability and sensitivity. Solutions derived for determining the stress level of individual-related phenomenons using such soft-computing methodologies offer a high degree of sensitivity and specificity required over other heuristic prediction approaches.

Our contributions are summarized as follows:

- **Key Problem**. We study workers' stress prediction problem in the context of human resource management. Studying this problem is vital to manage workers' stress and improve enterprise human resource management.

- **Novel Method**. We propose a deep learning-based method for the workers' stress prediction problem. We develop two stress prediction models (i.e., stress classification model and stress regression model) and correspondingly design two neural network architectures. We train these two stress prediction models based on the workers' data (e.g., salary, working time, KPI).

- **Extensive Evaluation**. We evaluate our method over two real-world datasets about workers' stress levels (i.e., ESI and HAJP). The experimental results show that our method can effectively predict workers' stress levels with high accuracy.

The remainder of this paper is organized as the following 5 parts: "Related work", "Preliminary", "Method", "Experiments", and "Conclusion and future work". We review literature related to our work in part 1. In part 2, we briefly introduce the background of HRM problem and neural network approach. In part 3, we propose the deep learning-based predictor to predict employees' stress levels and design the architecture of neural network we use. In part 4, we

carry out the experiments of our proposed deep learning-based predictor for employees' stress level prediction over two real-world datasets: ESI and HAJP. Finally, we conclude this paper and discuss future work in part 5.

## Related work

To mitigate work-related disorders, stress detection gains increasing research attention. The basic existing approach is the questionnaire-based stress level detection (WSQ [10]). However, the questionnaire-based stress level detection is potentially subjective such that the conclusions may be inaccurate for the decision-making. Therefore, we propose the artificial neural networks (ANN)-based stress level detection approach and apply this approach to any objective worker's data, including but not limited to salary, working time, KPI. One should note that our ANN-based detection is essentially different from the questionnaire-based stress level detection. Our approach is more general due to its objective, quantitative and automated properties. Once our approach trains a general prediction model based on the collected data, it can be adaptive to predict the new instances (i.e., workers). In contrast, the questionnaire-based stress level detection is more specific and lacks adaptation to new instances. Besides, it requires tedious manual effort to collect data and analyze data. We summarize the difference of them in Table 1 below.

### Neural network

The neural network has attracted a large amount of interest for training the mathematical model of complex systems [11, 12], and extending this concept to engineering applications. Since that, a sequence of significant works on the neural network has been proposed. Long Short-Term Memory (LSTM) [13] was proposed to deal with the gradient related issues of Recurrent Neural Network (RNN) with remarkable success in different applications, as speech recognition [14], machine translation [15], fault forecasting [16] and forecasting [17]. Then, the convolutional neural network (CNN) has been proposed for extracting features adaptively and effectively in different research fields [18–20]. In particular, deep learning theory [21] provides new perspectives for the feature representation of complex systems. Multiphase flow research that deep learning provides a lot of excellent solutions for its complex challenges. Ghosh *et al.* [22] and Roman *et al.* [23] separately use the genetic algorithm and the neural networks to identify the flow features. Some researchers try to use deep learning methods to study the pattern recognition [24] for further prediction. Deep learning was used to diagnose liver disease and predict the potential of liver disease [25]. Guo *et al.* [26] use the deep learning algorithm to identify the stock patterns for assisting the expert to predict the trend of the target stock. Recently, Le *et al.* [27] was attempting to predicting for Acute kidney injury (AKI) up to 48 hours in advance of onset by using CNN and patient Electronic Health Record (EHR) data, in order to overcome the difficulties of prediction and diagnosis.

Here, we can conclude that deep learning algorithms have achieved a giant success in numerous research fields along with impressive and profound impacts in real-world

**Table 1. Comparing our method with questionnaire-based stress level detection (WSQ).**

| Ours | WSQ [10] |
|---|---|
| General | Specific |
| Adaptive | Static |
| Quantitative | Qualitative |
| Automatic | Manual |

applications. The outstanding performance in industry, economy and medical science, indicates the potential of management.

## Neural network-based stress level analysis

In the next part, we introduce some works related to stress analysis based on the neural network. Singh *et al.* [28] investigate continuous stress level monitoring based on neural network classifier. The features they use are physio-logical signals like Galvanic Skin Response (GSR) and Photoplethysmography (PPG) and the network architecture they design is Layer Recurrent Neural Networks (L-RNN). However, it is not easy to obtain physio-logical signals. Requiring the workers to wear the handsets is intrusive. Mohanty *et al.* [29] proposes a feature reduction mechanism using the combination of Vector Quantization (VQ) and eigenvalue decomposition and select effective features to recognize child emotion using the probabilistic neural network. Lin *et al.* [30] study psychological stress detection from cross-media micro-blog data using deep sparse neural network. Weng *et al.* [31] train different types of classifiers using the neural network to predict the disease. The existing works [32–34] are most related to our method. Umematsu *et al.* [32] study students' daily life stress forecasting based on the long short-term memory (LSTM) neural network and students' data (i.e., physiological signals, mobile phone usage, location, and behavioral surveys). Differently, we study the workers' stress prediction problem in the context of human resource management and use the enterprise-related datasets. Li *et al.* [33] study stress detection using 1-dimensional (1D) convolutional neural network and a multilayer perceptron neural network. Note that the dataset they use is physiological data collected from chest-worn and wrist-worn sensors. Collecting the physiological data from workers is difficult due to requiring the workers to wear the handset sensors. Bekesiene *et al.* [34] propose an ANN-based framework to predicate the stress level of military conscripts. However, they use the military data, which is secret. Besides, they do not study the stress regression model, which is one of focus in our work.

## Preliminary

### Constitutive definitions of HRM policies

In this part, we review the constitutive definitions of typical HRM policies [35] as Table 2. Some research points out that perceptions of HRM are associated with levels of employee well-being [36], especially the emotional factors. The results show that the relationship between work motivation and the level of stress, is positive, which leads to the improvement of

**Table 2. The constitutive definitions of HRM policies.**

| HRM Policy | Constitutive Definition |
|---|---|
| Recruitment and Selection (RS) | Look for and select employees, in terms of: people's values, interests, expectations. on demands of the position. |
| Involvement (I) | Contribute to employees' well-being, in terms of: acknowledgement, relationship, participation and communication. |
| Training, Development & Education (TD&E) | Provide employees systematic competence and stimulate continuous learning and knowledge. |
| Work Conditions (WC) | Provide good work conditions to employees, in terms of: benefits, health, safety and technology. |
| Competency-Based Performance Appraisal (CBPA) | Evaluate employee's performance for decisions on promotions, career planning and development. |
| Compensation and Rewards (CR) | Reward employees' performance and competence via remuneration and incentives. |

employee's performance straightly. Therefore, in our work, the research focuses on involvement rather than other HRM policies.

## The introduction of neural network

In this part, we review the technique of deep neural network (i.e., DNN), which serves as the basis of our work. DNN consists of one input layer, numerous hidden layers, and one output layer. In each layer, there are some neurons, of which the link is a cause-effect chain to be learned and trained. Due to the non-linear representation ability, DNN is helpful to uncover the underlying non-linear relation in the datasets. In our setting, the feature of the worker is characterized by a 1-dimensional vector. Therefore, the fully connected network (also known as Multi-layer perceptron, MLP) is applicable in our setting. Note that we do not use a convolution neural network (CNN) because the datasets for workers' stress prediction do not often contain 2-dimensional image feature like MNIST.

The fully connected layer indicates that a neuron is connected to all neurons in the next layer. The DNN model is of a linear function and an activation function as Eq 1,

$$a = \sum w_i x_i + b_i, \tag{1}$$

where $x_i$ is the input value of every neuron; $w_i$ is the coefficient of linear relationship, and $b_i$ is the bias.

Assuming there are $L$ hidden layers in DNN, we use $f(x)$ to represent the activation function for increasing the nonlinearity of the neural network, which is usually applied for evaluating the nonlinear function for nonlinear models. The calculation of the output value can be expressed as Eq 2:

$$\begin{aligned} f(x) = & \ f[a^{L+1}(h^L(a^L(\ldots(h^2(a^2(h^1(a^1(x))))))))], \\ a^L(x) = & \ W^L x + b, \end{aligned} \tag{2}$$

where $L$ represents the $L$-th layer; $x$ is the matrix of input variables; $W$ and $b$ are high dimensional matrices.

The loss function is used to estimate the difference between the real value and the predicted value. The values are non-negatively real, which is the baton of the whole network model. They guide the network parameter learning through the error backpropagation generated by the predicted sample and the real sample marks. When the value of the loss function is smaller, the robustness of the model is better. $W$ and $b$ are determined by the minimum value of the loss function in the process of training. The last layer of the network has only one neuron of *LI*. Thus, the Binary_Crossentropy loss function was used to train the network, instead of using the conventional softmax classification loss function. For all $N$ points,

$$H_p(q) = -\frac{1}{N} \sum y_i \cdot \log(p(y_i)) + (1 - y_i) \cdot \log(1 - p(y_i)), \tag{3}$$

where $y_i$ is the label (1 for positive point and 0 for negative point); $p(y)$ is the predicted probability of the point to be positive.

## Method

We propose a deep learning-based predictor to predict employees' stress levels. The employee's feature is represented by a vector with predefined dimensions. We construct the neural network architecture using Multi-layer perceptron (MLP), which is essentially a fully connected network. In our problem, there are two kinds of output layers. When the prediction is binary

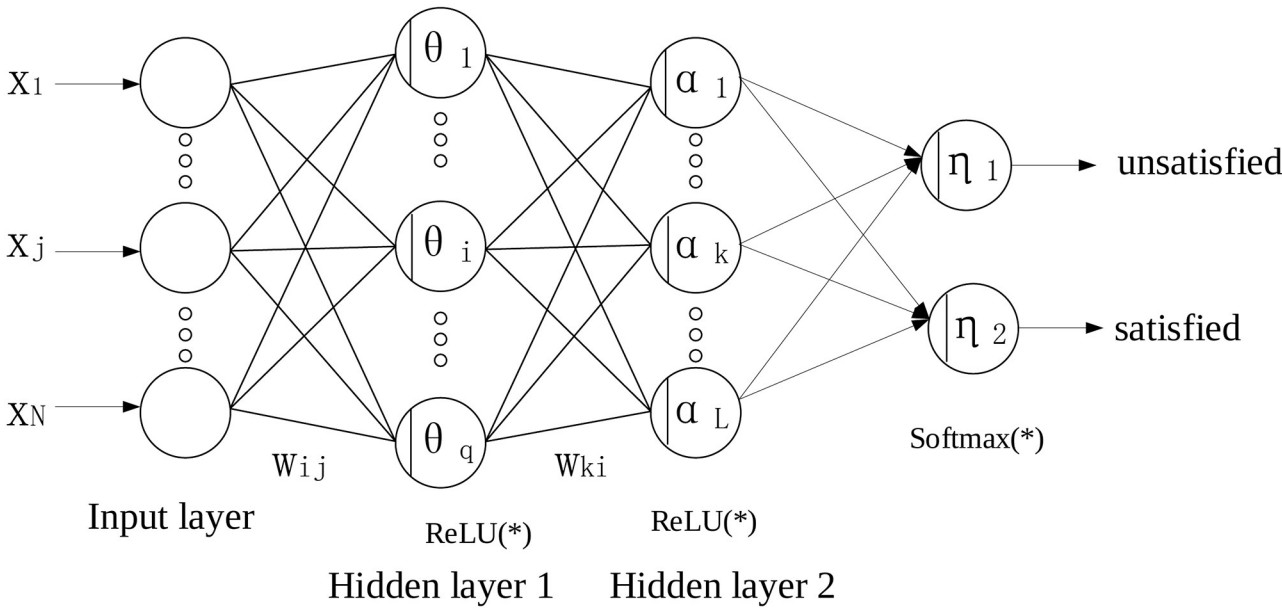

**Fig 1. The network architecture of the classification model used in our setting.**

results that characterize workers' satisfied statuses or unsatisfied statuses, the output layer consists of 2 neurons. When the prediction is the numeric result that characterizes workers' satisfactory level in the interval [0, 1], the output layer consists of 1 neuron. And these two networks are called the classification model and regression model, respectively. We illustrate the network architectures of the stress classification model and stress regression model in Figs 1 and 2, respectively. In Fig 1, the input layer consists of several neutrons that correspond to the number of worker's features. Following the input layer, several hidden layers are set up to

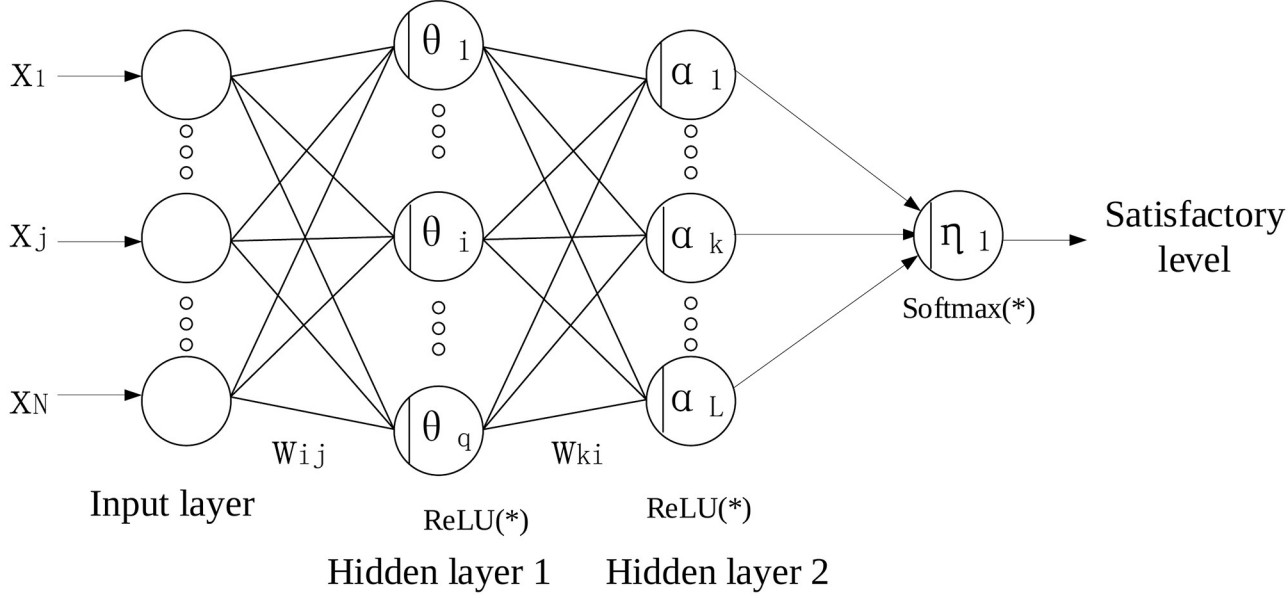

**Fig 2. The network architecture of the regression model used in our setting.**

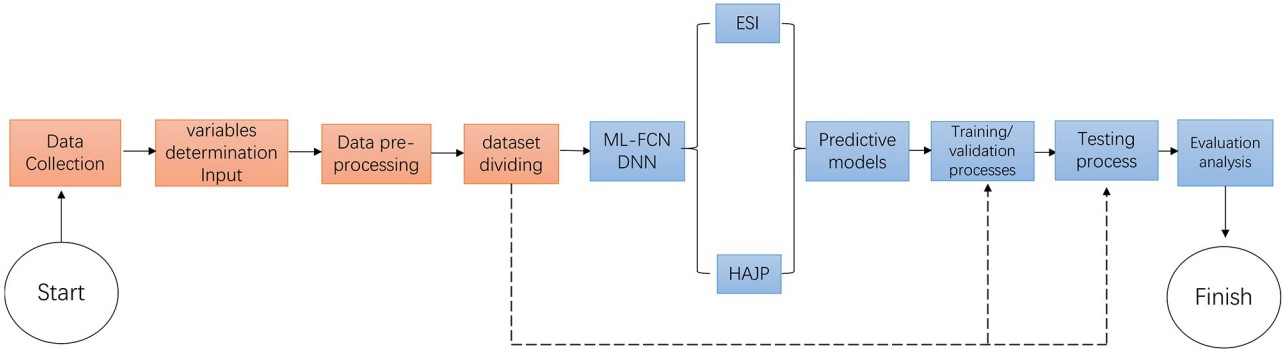

**Fig 3. The procedure of constructing a predictive model.**

extract the worker's feature in the more abstract representation spaces layer by layer. Due to the binary classification in the stress classification model, we set up 2 neutrons in the output layer. Similar network architecture is used in the stress regression model as shown in Fig 2. Differently, the output layer consists of only one neutron due to the scalar stress level prediction in the stress regression model. Actually, we can construct the stress regression model by adding a neutron as the output layer in the stress classification model.

By setting up the network architectures, we can formally develop our method for the workers' stress prediction problem. Taking two real-world datasets (ESI for the stress classification model and HAJP in the stress regression model) as examples, we illustrate our method in Fig 3. The procedure of constructing a predictive model can be summarized as follows: 1) Data collection: system receives the datasets (e.g., ESI and HAJP) as inputs; 2) Variables determination input: System identifies the variables and inputs the determined variables; 3) Data preprocessing: Before model training, system scrambles and generalizes the variables due to a different physical unit of the variables; 4) Database dividing: System divides the dataset into three shares for model training, model validation, and model testing; 5) ML-FCN DNN: System runs DNN over the training set and checks the accuracy of the training DNN model over the validation set. When the training process is finished, the system uses the test set to evaluate the generalization ability of the model and further validate the prediction accuracy.

## Experiments

### Setup

We carry out experiments to evaluate the proposed predictor over two datasets: Employee Satisfaction Index (ESI) https://www.kaggle.com/mohamedharris/employee-satisfaction-index-dataset and Hr Analytics Job Prediction (HAJP) https://www.kaggle.com/mfaisalqureshi/hr-analytics-and-job-prediction, which are publicized in Kaggle.

In ESI, there are 500 workers with 14 attributes. We exclude the index id and worker id and set the valid features as: "age, dept, location, education, recruitment_type, job_level, rating, onsite, awards, certifications, and salary". Among these features, the numeric ones include "age, job_level, rating, onsite, awards, certifications and salary". For attributes "job_level and rating", their values are ranged in {1, 2, 3, 4, 5}. For attribute "dept", it represents the department worker belongs to and its domain is {HR, Technology, Sales, Purchasing, Marketing}. For attribute "education", it represents the degree worker has and its domain includes undergraduate (UG) and postgraduate (PG). For attribute "location", it represents the location worker lives and its domain includes city and suburb. For attribute "recruitment_type", its

domain is {Referral, Walk- in, On- Campus, Recruitment Agency}. The label is "satisfied" which is a binary attribute. Thus, our predictor for the ESI dataset belongs to the classifier model.

In HAJP, there are 14999 workers with 10 attributes. We select "last_evaluation, number_project, average_montly_hours, time_spend_company, Work_accident, left, promotion_last_5years, Department, salary" as valid features. The label is "satisfaction_level" which is floating attribute and ranged in [0, 1]. Thus, our predictor for the HAJP dataset belongs to the regression model.

We implement the proposed predictor using the Pytorch framework with version 1.6.0. We run all the experiments on a Ubuntu 16.04.3 server with five NVIDIA TITAN xp graphic cards. We set up 2 hidden layers for our MLP-based predictor. We split the whole datasets into training datasets, validating datasets, and testing datasets with a ratio 6: 2: 2. The maximum training steps are set to be 500 by default. To conveniently demonstrate the performance of testing phase, we report the test results during the training process.

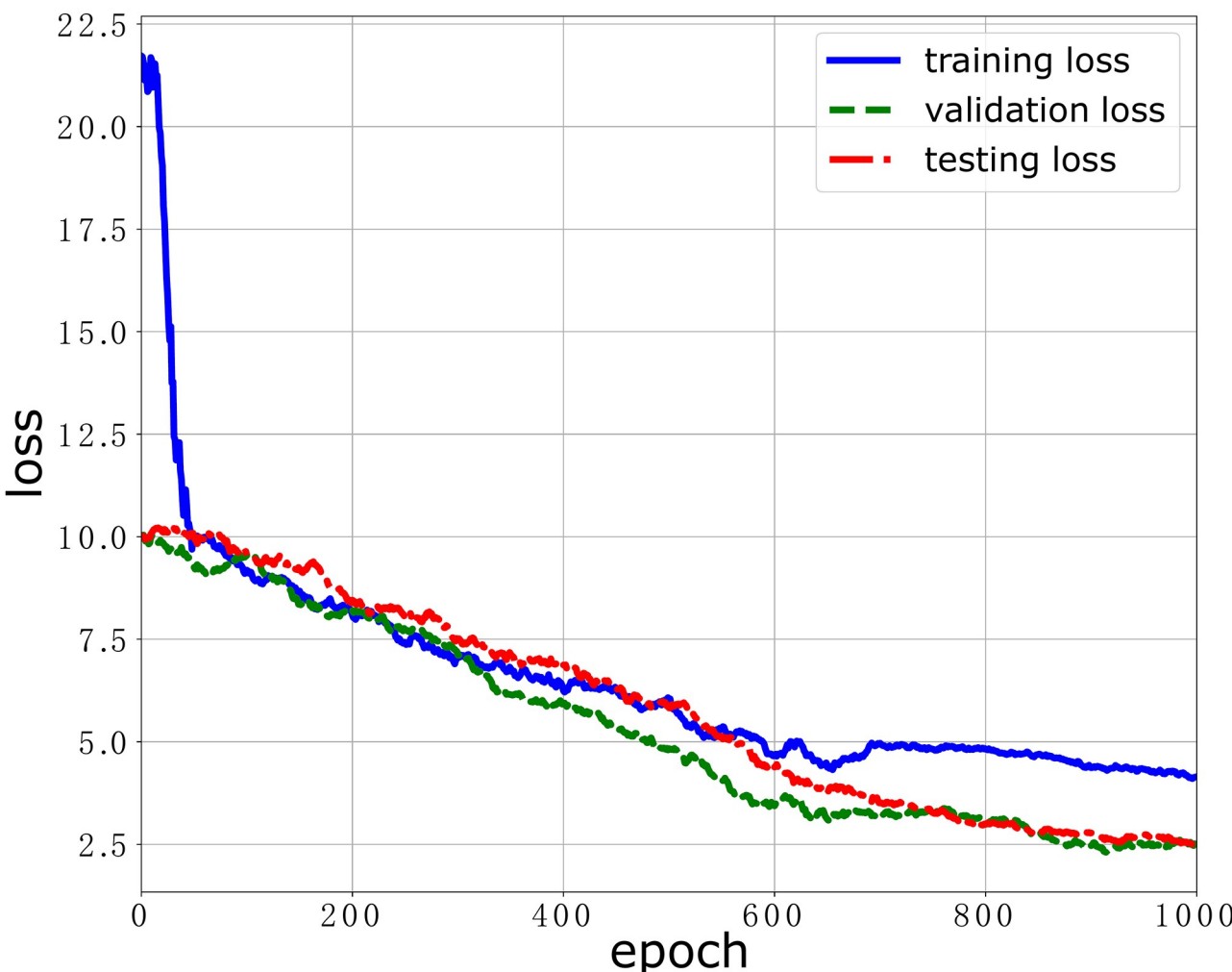

**Fig 4. The model loss performance of our DNN model over the ESI dataset.**

## Numerical results

**The performance of stress classification model.** We first evaluate the performance of our stress classification model over the ESI dataset where the labels are the binary attribute: satisfied and unsatisfied. The performance indicators to evaluate the stress classification model include model loss and model accuracy. We train the stress classification model up to 1000 epochs and evaluate the model loss and model accuracy in the training stage, validation stage and testing stage, respectively. The results are plotted in Figs 4 and 5. As shown in Fig 4, we can see that the model losses in the training stage, validation stage and testing stage decrease with the training step. When the training step is close to 600, the loss function gradually converges. This means that the model can learn to predict the workers' stress from the training dataset as long as enough training steps are taken. As shown in Fig 5, we can also see that the model accuracies in the training stage, validation stage and testing stage increase with the training step. After 200 epoch, the model accuracy of 3 curves for the training stage, validation stage and testing stage becomes stable and reaches the peak value. This also validates that the stress classification model can predict accurately workers' stress. The

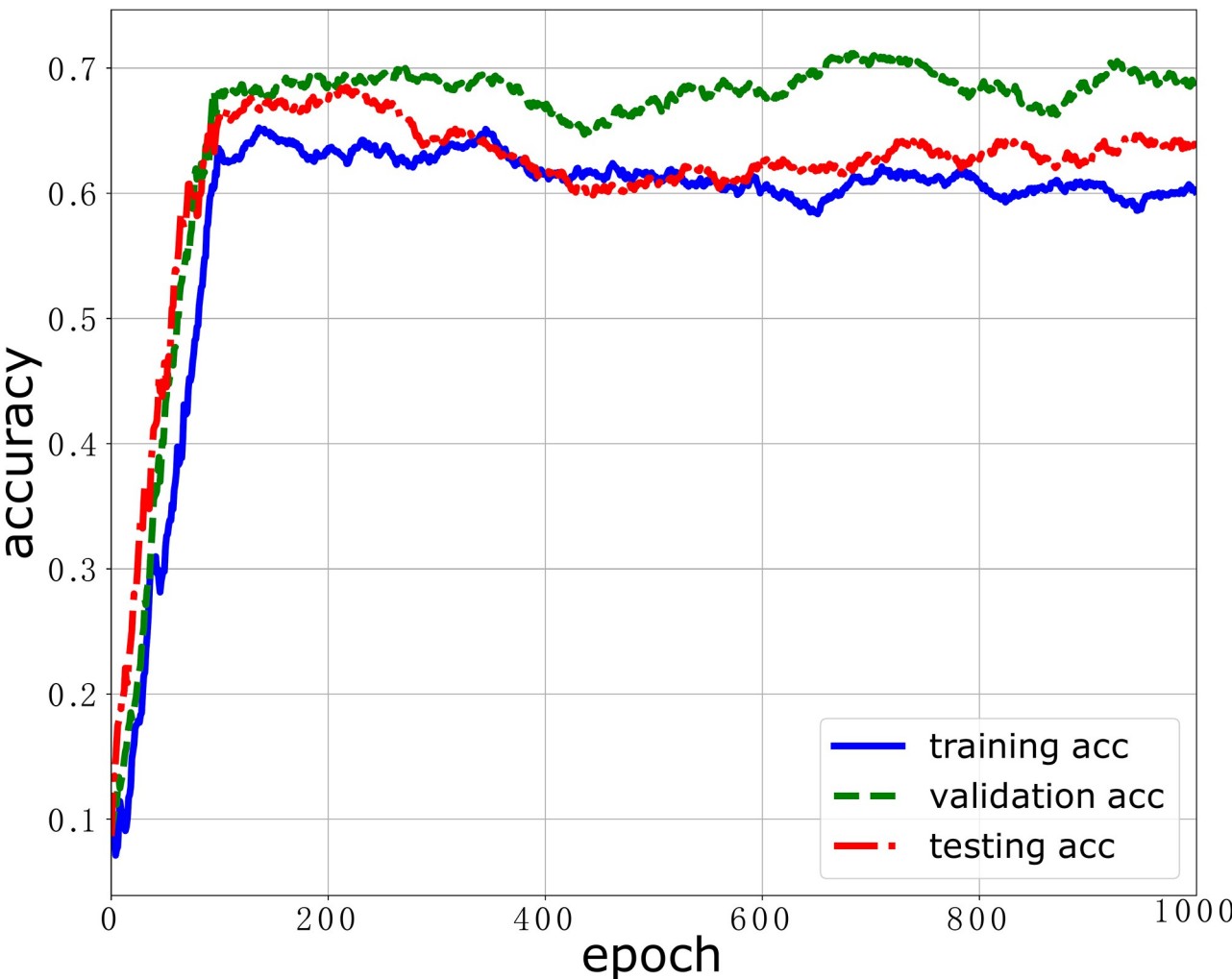

**Fig 5. The model accuracy performance of our DNN model over the ESI dataset.**

**Table 3. Comparing our stress classification model with SVM-based stress classifier.**

| Methods | Ours | SVM |
|---|---|---|
| Accuracy | 71.2% | 60.9% |
| F1 score | 68.6 | 57.9 |

maximum accuracy our stress classification model achieves in the validation stage is 71.2%. The model performance can be improved when the samples are heterogeneous and the size of the training dataset is increased. When the data samples are more diverse and heterogeneous, the data distribution in the dataset is more approximate to the original ground-truth distribution. Thus, the trained model can learn more and have a more powerful prediction. And when the size of dataset is large, the dataset is more likely diverse and heterogeneous [37, 38].

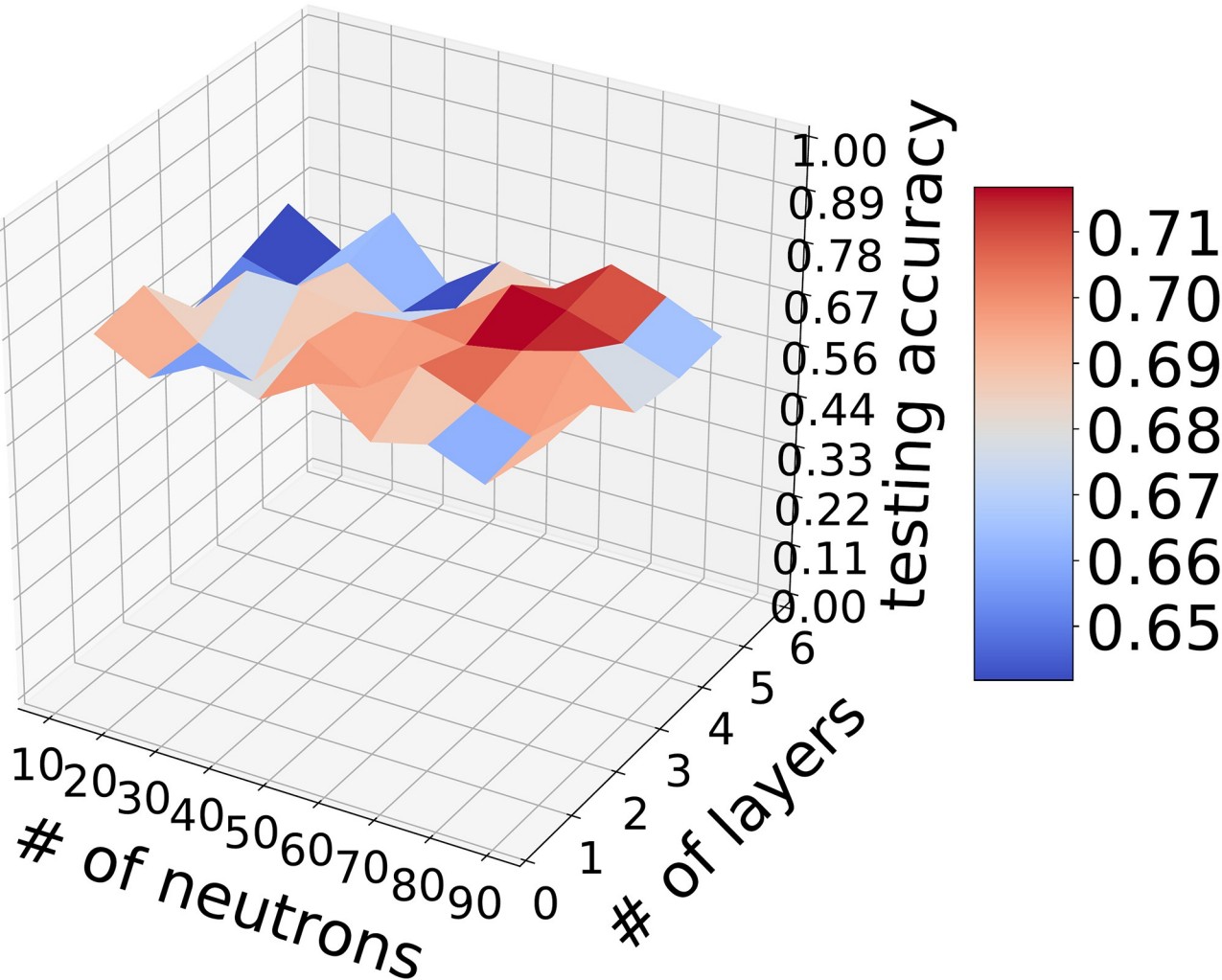

**Fig 6. The performance of DNN's architecture.**

It should be noted that the test data was never used in the training phase though we report the testing result in each time step in Figs 4 and 5.

*The comparison with SVM classifier.* To verify the performance of our stress classification model, we compare our stress classification model with a baseline: SVM-based stress classifier, a traditional classification approach. We still use the same experimental setup: train the models 1000 epochs and report the accuracy and F1 score results. We summarize the comparison result in Table 3.

From Table 3, we can see that the accuracy and F1 score metrics of our stress classification model are 71.2% and 68.6, respectively. The accuracy and F1 score metrics of the SVM-based stress classifier are 60.9% and 57.9, respectively. Therefore, our stress classification model outperforms the SVM-based stress classifier when predicting workers' stress statuses.

*The impact of number of layers and neurons.* Next, we evaluate the impact of the network's architecture (i.e., the number of the network's layers and neurons) on the model accuracy. We vary the number of total neurons in {10, 20, . . ., 90} and the total hidden layers in {1, 2, . . ., 6}.

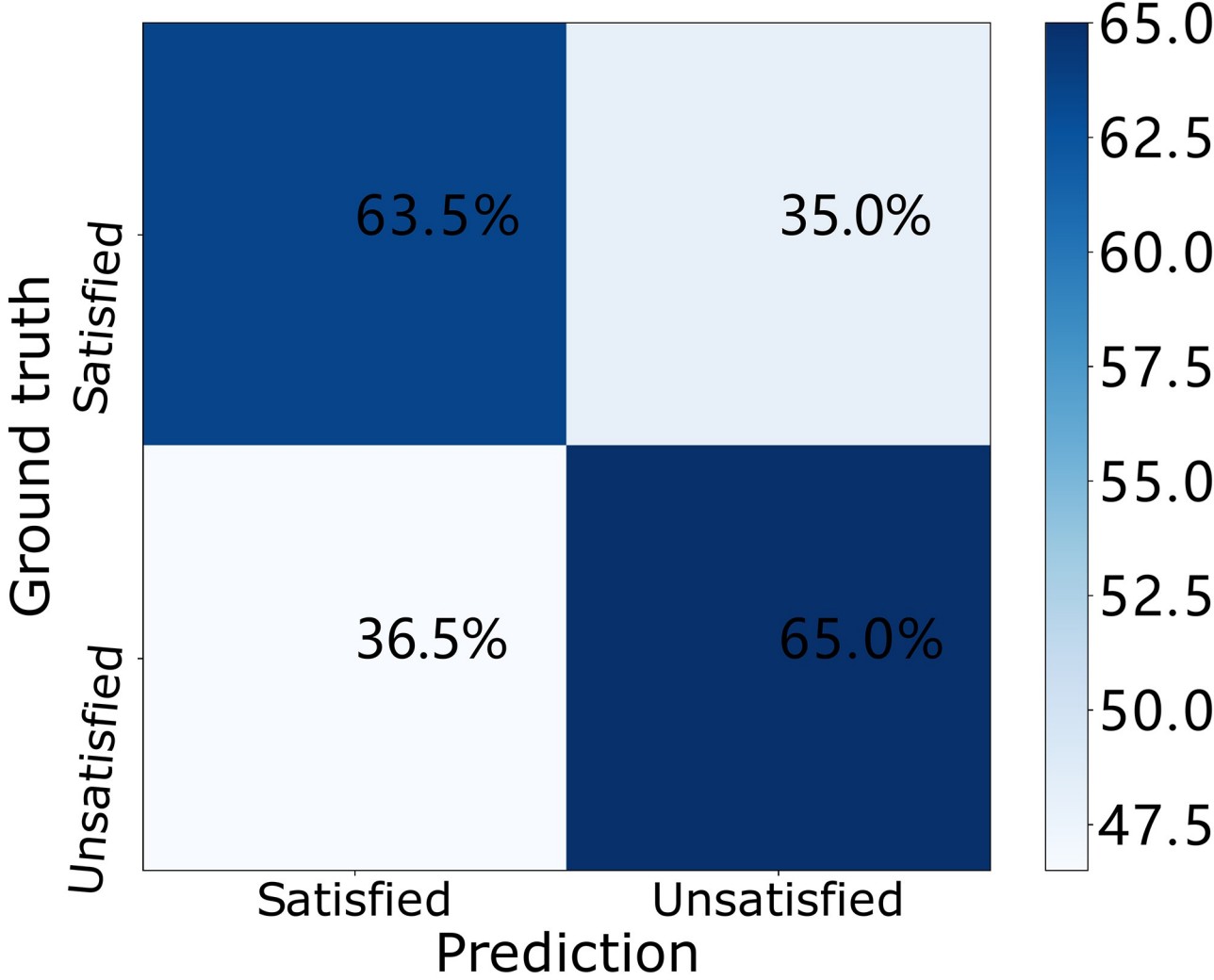

**Fig 7. Confusion matrices of our DNN model over the ESI dataset in the training phase.**

We plot the result in Fig 6. As shown in Fig 6, we can see that the testing accuracy has significant non-convex relations with the number of the network's layers and neurons. This means that it is difficult to optimize the hyperparameters (i.e., the number of the network's layers and neurons) in DNN. Roughly speaking, the optimal accuracy 71.2% can be achieved when the number of the network's neurons is set to be 50 and the number of the network's layers is set to be 5. It is possible that there exists a global maximum with respect to the number of the network's layers and neurons. Applying the hyperparameter optimization technique, the optimal hyperparameters can be determined. However, it would be expensive to find the optimal hyperparameters due to the difficulty of non-convex optimization. And we will choose the near-optimal hyperparameters (i.e., 50 neurons and 5 hidden layers) in our paper.

To further investigate the classification performance of our stress classification model, we collect the accuracy results in the training stage, validation stage and testing stage and plot the results in the confusion matrix in Figs 7–9. In each confusion matrix, each row represents ground truth labels (stress status) while each column corresponds to predicted labels. In the training phase (i.e., Fig 7), 63.5% workers are recognized as "satisfied" while their actual stress statuses are "satisfied". And 65.0% workers are recognized as "unsatisfied" while their actual stress statuses are "unsatisfied". In the validation phase (i.e., Fig 8), 71.2% workers are

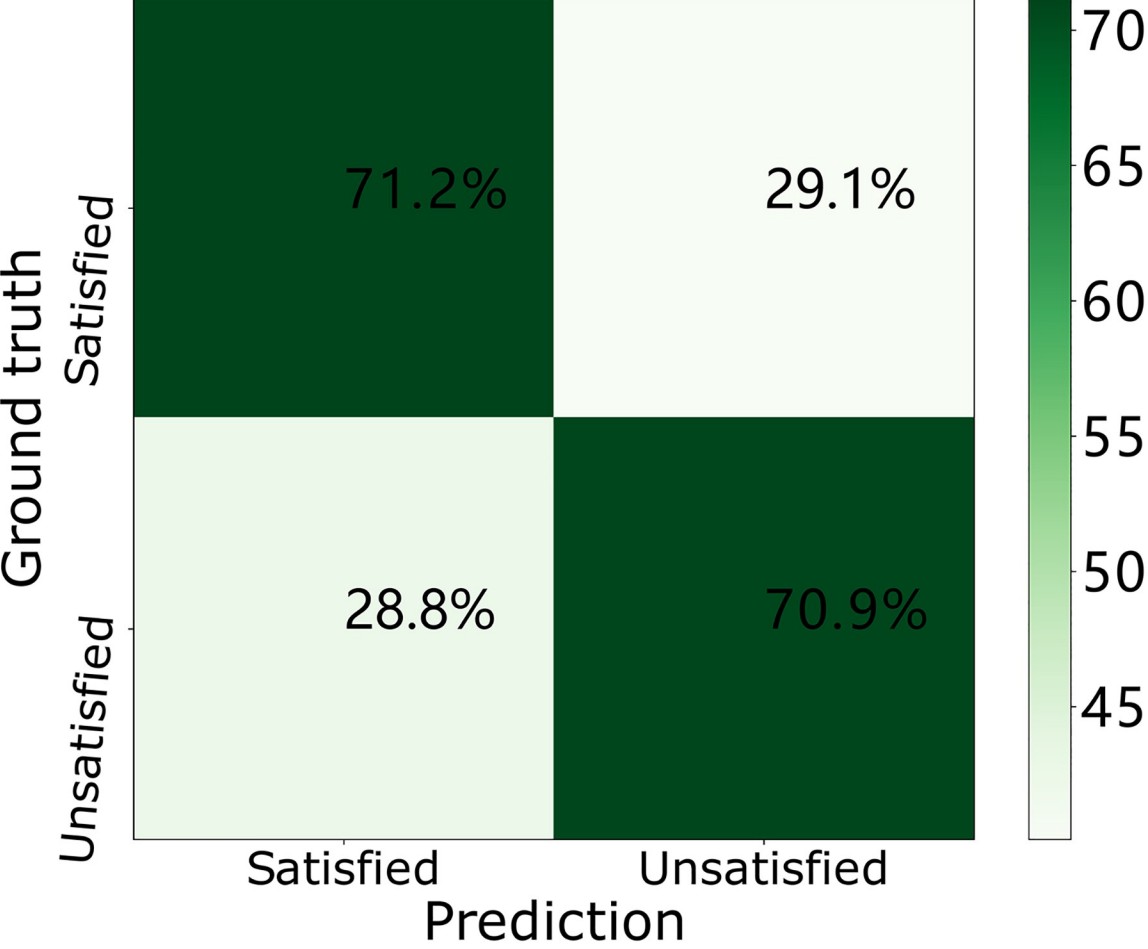

**Fig 8. Confusion matrices of our DNN model over the ESI dataset in the validation phase.**

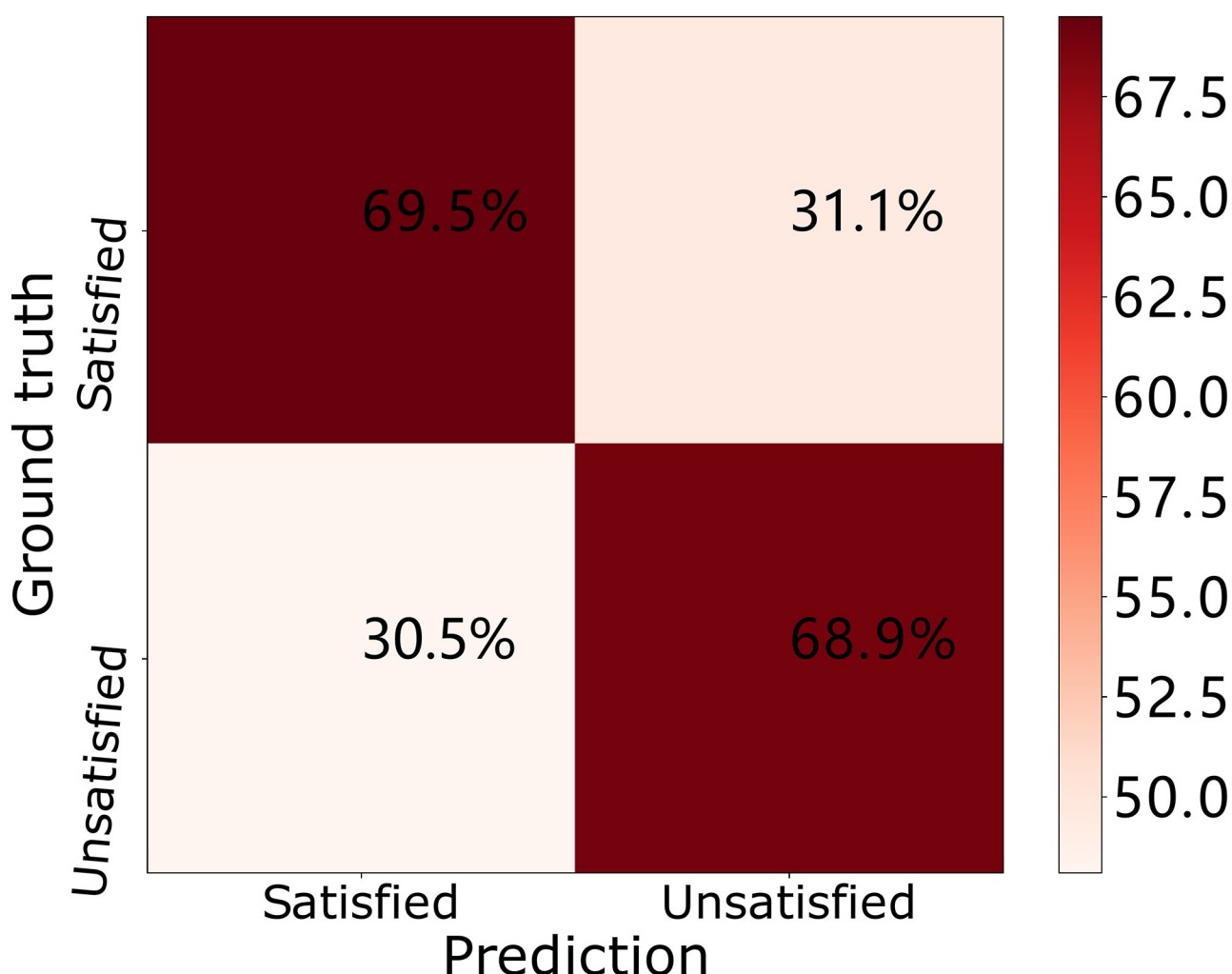

**Fig 9. Confusion matrices of our DNN model over the ESI dataset in the testing phase.**

recognized as "satisfied" while their actual stress statuses are "satisfied". And 70.9% workers are recognized as "unsatisfied" while their actual stress statuses are "unsatisfied". In the testing phase (i.e., Fig 9), 69.5% workers are recognized as "satisfied" while their actual stress statuses are "satisfied". And 68.9% workers are recognized as "unsatisfied" while their actual stress statuses are "unsatisfied".

**The performance of stress regression model.** Next, we evaluate the performance of our stress regression model over the HAJP dataset where the labels are the numeric attribute in the interval [0, 1]. The difference between the stress classification model and the stress regression model is that the stress classification model has two neutrons in the output layer while the stress regression model has only one neutron in the output layer. This is because the label in the stress classification model is binary while the label in the stress classification model is numeric. Thus, the performance indicator for evaluating the stress classification model is model loss. We plot the results in Figs 10 and 11. As shown in Fig 10, we can see that the model losses for the stress classification model in the training stage, validation stage and testing stage decrease with the training step. When the training step is close to 650, the loss function gradually converges. This means that the model can learn to fit the workers' stress level from

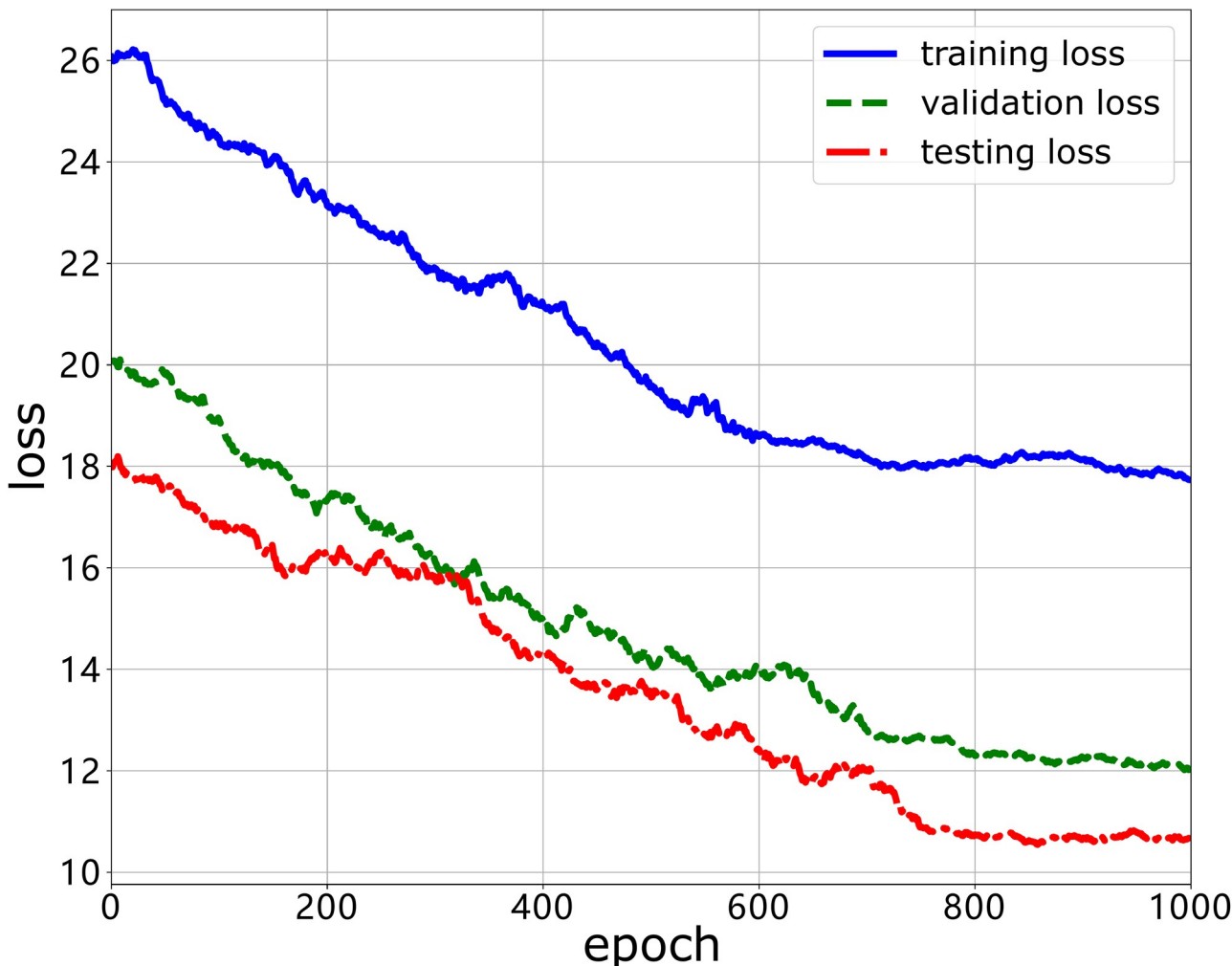

**Fig 10. The model loss performance of our DNN model over the HAJP dataset.**

the training dataset (i.e., HAJP) as long as enough training steps are taken. In Fig 11, we evaluate the impact of the selected features on the model loss reduction. We can see that the model loss is reduced significantly when a large number of features is selected to train the stress regression model. However, the more features are used to train the model, the marginal model loss is reduced less. This demonstrates the diminishing return property in the relationship between model loss and the number of features.

## Conclusion and future work

In this paper, we investigate a DNN-based stress predictor for HRM, in terms of involvement; training, development and education; work conditions; and compensation and rewards. Our work fills the gap in HRM by introducing a scientific measurement to employees' stress levels, predicting the potential mental issues and assisting the manager to conduct the effective measure. In the experiments, the first test is in the ESI dataset of 500 workers with 14 attributes. Then, in the HAJP dataset, the training and validating datasets are extended to be of 14999 workers with 10 attributes to build our regression model on stress level. The experimental

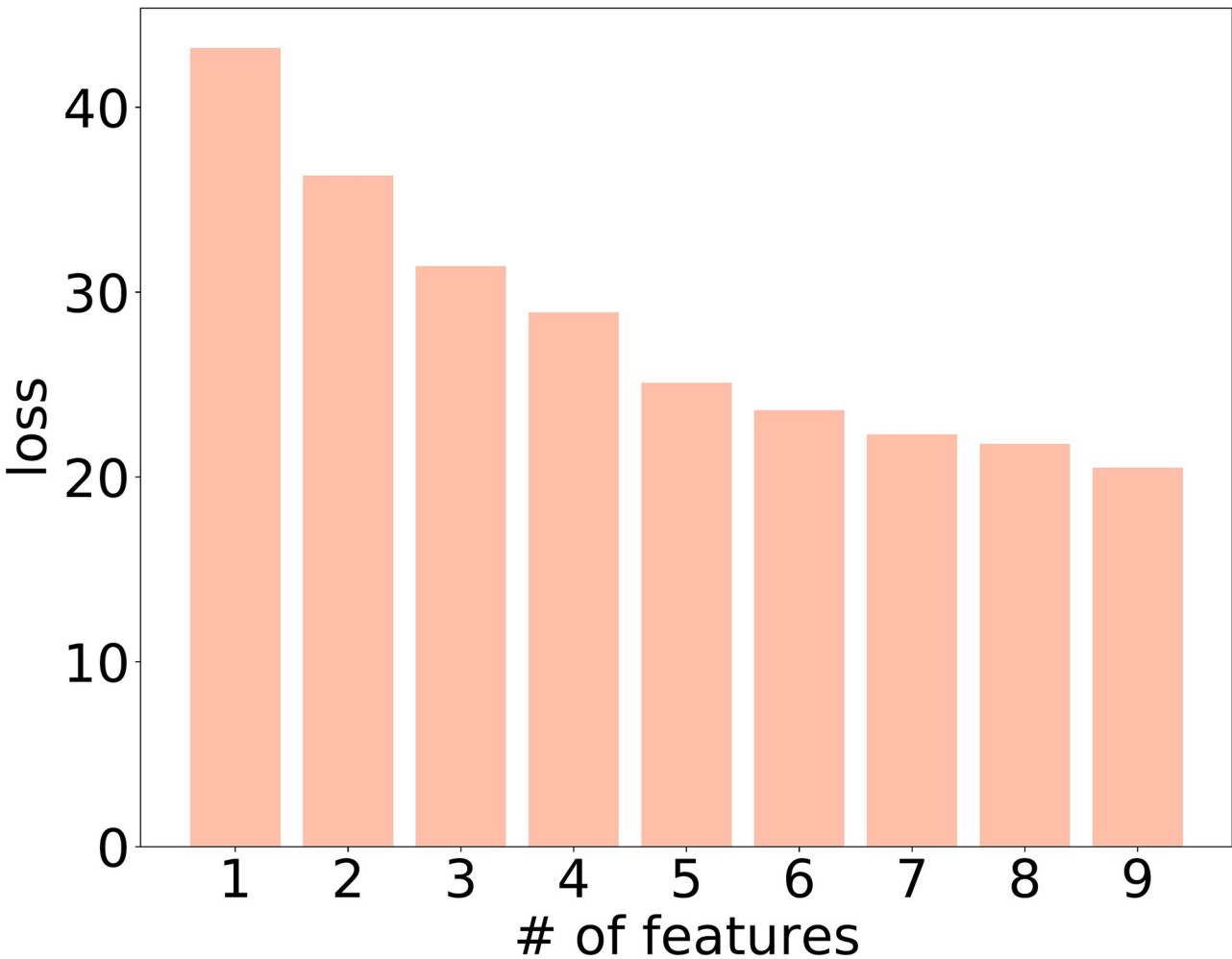

**Fig 11. The impact of # of features on loss reduction in the HAJP dataset.**

results show that our proposed deep learning based approach can effectively predict workers' stress status with 71.2% accuracy in the classification model and 11.1 prediction loss in the regression model. By accurately predicting workers' stress status with our method, the HRM of enterprises can be improved.

Inspired by our stress predictor, there are some interesting issues to be exploited. First, is there any scientific decision maker to solve the predicted mental issues? Then, there exist other mental diseases which will make an influence on working performance, beyond the stress-related mental diseases. We should attempt to find a more comprehensive mental disease predictor with high accuracy. Finally, the prediction and decision-making assistance should be of more crucial issues rather than individuals' issues, such as the development policies and marketing strategies of enterprises.

## Supporting information

**S1 File. ESI dataset.**
(CSV)

**S2 File. HAJP dataset.**
(CSV)

## Acknowledgments

The authors would like to thank all the editor and reviewers for their valuable comments.

## Author Contributions

**Conceptualization:** Yu Zhang.

**Data curation:** Yu Zhang.

**Formal analysis:** Yu Zhang.

**Investigation:** Yu Zhang.

**Methodology:** Yu Zhang.

**Software:** Yu Zhang.

**Supervision:** Ershi Qi.

**Validation:** Yu Zhang.

**Visualization:** Yu Zhang.

**Writing – original draft:** Yu Zhang.

**Writing – review & editing:** Ershi Qi.

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
