## [Decision Letter · Decision Letter 0]

29 Dec 2021

PONE-D-21-32032Happy work: Improving enterprise human resource management by predicting workers' stress using deep learningPLOS ONE

Dear Dr. zhang,

Thank you for submitting your manuscript to PLOS ONE. After careful consideration, we feel that it has merit but does not fully meet PLOS ONE’s publication criteria as it currently stands. Therefore, we invite you to submit a revised version of the manuscript that addresses the points raised during the review process.

We look forward to receiving your revised manuscript.

Kind regards,

Ashraf Imran, Ph.D.

Academic Editor

PLOS ONE

Journal Requirements:

2. Please amend the manuscript submission data (via Edit Submission) to include author Hongwei Liu.

3. Please amend your authorship list in your manuscript file to include author Ershi Qi.

Additional Editor Comments:

Based on the reviewer's comments, the authors are advised to make "major revisions" and resubmit the revised version. Authors are also advised to take special care of English typos and errors.

Reviewers' comments:

Reviewer's Responses to Questions

**Comments to the Author**

1. Is the manuscript technically sound, and do the data support the conclusions?

Reviewer #1: Partly

Reviewer #2: Yes

Reviewer #3: Yes

2. Has the statistical analysis been performed appropriately and rigorously? 

Reviewer #1: Yes

Reviewer #2: N/A

Reviewer #3: Yes

3. Have the authors made all data underlying the findings in their manuscript fully available?

Reviewer #1: Yes

Reviewer #2: Yes

Reviewer #3: Yes

4. Is the manuscript presented in an intelligible fashion and written in standard English?

Reviewer #1: Yes

Reviewer #2: Yes

Reviewer #3: Yes

5. Review Comments to the Author

Reviewer #1: - The idea of detecting stress level based on factual data rather than subjective data with ANNs is novel. However, the manuscript should be organized better to support this proposal.

- The work proposes detecting stress level of workers based on objective worker’s data would deliver better results than a potentially subjective questionnaire. After that it conducts experiments on some objective data of workers and finds accuracy results. However, merely delivering this result doesn’t support its claimed novelty; therefore, a comparison between stress level detection accuracy based on a questionnaire and based on proposed method should be illustrated.

- The performance of proposed model (using ANNs for classification) may be compared to performance of other non-SOTA ML classifiers (i.e. SVM) to see its contribution.

- Access links of the datasets are not available via Kaggle but they are attached to the manuscript. Therefore, it is not clear that these are existing sets that developed by others or newly developed by the authors. If the data is developed by the authors, then data generation process should be illustrated. The Figure 2 gives steps about data collection and preprocessing, the manuscript should also detail these steps.

- In Figure 5, the confusion matrices are given based on percentages of total number of workers in ESI dataset (i.e., 500 workers). When the given percentages are multiplied with the total worker number, the sum doesn’t end up to 500. In other words, the percentages do not sum up to 1. Therefore, it is not possible to compute the accuracy metric from the confusion matrix.

- In Figure 2, testing process is shown as a distinct process then the training process. The validation process is not shown in the figure; however, it is expected to be part of the training process. The accuracy metric of the manuscript is shown as performance of the validation process, which is not expected to be. The overall performance of the network should be indicated as performance of a distinct testing process on never-seen data. This should clearly be illustrated in the Figure 2 and results should be organized accordingly.

- The performance metric (i.e. accuracy) should be defined clearly and results should accompany it.

- The manuscript includes some capitalization and grammatical errors.

- In the Related Works section, the general review of ANN’s is given lengthy compared to works related to stress level detection methods. When a quick search is conducted, it is possible to find various methods for stress detection with ANN based on different type of data. A couple of them are:

o https://www.mdpi.com/2227-7390/9/6/626/htm

o https://bmcmedinformdecismak.biomedcentral.com/articles/10.1186/s12911-020-01299-4

o https://ieeexplore.ieee.org/abstract/document/8834624

- This statement is not tested, or results are not illustrated: “The model performance can be improved when the size of the training dataset is increased and the samples are heterogeneous.”

- In Figure 3, the given training, validation, and test set graphs imply that they are generated during the training process. In other words, the test data appears to be used in the training process, however, it is possible that the corresponding intermediate results on the test data did not contribute to the training process contrary to validation results. If this is the case (i.e. test data is tested on the intermediate trained model but the results were not used to improve the intermediate model), then it should clearly be indicated in the manuscript. If the test data was never used during the training, then the figures should be explained accordingly.

Reviewer #2: 1- it is recommended to add the structure of the paper at the end of introduction section

2-contribution 2 is not clear which is the main contribution of this study author should explain it in details

3-the results are not discussed, need to discus results critically

4- author could use the recent publish work in Deep Learning application such as

https://journal.esj.edu.iq/index.php/IJCM/article/view/100

Reviewer #3: Thanks to the authors for this research effort and the idea that contributes to promoting an intelligent, AI-based employee management approach, specifically deep learning, away from traditional methods. The research is interesting in its idea, implementation and depiction of its results.

6. PLOS authors have the option to publish the peer review history of their article (what does this mean?). If published, this will include your full peer review and any attached files.

Reviewer #1: No

Reviewer #2: No

Reviewer #3: No

---

## [Author Response · Author response to Decision Letter 0]

25 Feb 2022

Response to Reviewers’ Comments

We would like to thank the editor and the reviewers for their careful reading and valuable comments that have helped us to improve this manuscript. We have made several improvements to the manuscript following the reviewers’ comments and suggestions. We highlighted the main changes in the draft in blue for the reviewers’ convenience. Now we first summarize these main changes and then respond to each of the reviewer’s comments in detail.

1. We have discussed the difference between our work and the questionnaire based stress level detection (Work Stress Questionnaire, WSQ [1]).

2. We have added three most related works and described their difference from our work to highlight the significance.

3. We have corrected the figures, typos and confusions according to reviewers’ comments.

As for the detailed responses to each comment, please refer to our response letter.

---

## [Decision Letter · Decision Letter 1]

21 Mar 2022

Happy work: Improving enterprise human resource management by predicting workers' stress using deep learning

PONE-D-21-32032R1

Dear Dr. zhang,

We’re pleased to inform you that your manuscript has been judged scientifically suitable for publication and will be formally accepted for publication once it meets all outstanding technical requirements.

Kind regards,

Ashraf Imran, Ph.D.

Academic Editor

PLOS ONE

Additional Editor Comments (optional):

Reviewers' comments:

Reviewer's Responses to Questions

**Comments to the Author**

1. If the authors have adequately addressed your comments raised in a previous round of review and you feel that this manuscript is now acceptable for publication, you may indicate that here to bypass the “Comments to the Author” section, enter your conflict of interest statement in the “Confidential to Editor” section, and submit your "Accept" recommendation.

Reviewer #1: All comments have been addressed

Reviewer #2: All comments have been addressed

2. Is the manuscript technically sound, and do the data support the conclusions?

Reviewer #1: Yes

Reviewer #2: Yes

3. Has the statistical analysis been performed appropriately and rigorously? 

Reviewer #1: Yes

Reviewer #2: Yes

4. Have the authors made all data underlying the findings in their manuscript fully available?

Reviewer #1: Yes

Reviewer #2: Yes

5. Is the manuscript presented in an intelligible fashion and written in standard English?

Reviewer #1: Yes

Reviewer #2: Yes

6. Review Comments to the Author

Reviewer #1: Revised version is well-worked and now I believe ready for publication. Congratulation to the authors and thanks for their contribution to the field!

Reviewer #2: Paper could be accepted and authors addressed all reviewers comments.the paper is good and could be useful for many companies

7. PLOS authors have the option to publish the peer review history of their article (what does this mean?). If published, this will include your full peer review and any attached files.

Reviewer #1: No

Reviewer #2: **Yes: **Mohammad Aljanabi

---

## [Editor Report · Acceptance letter]

1 Apr 2022

PONE-D-21-32032R1 

Happy work: Improving enterprise human resource management by predicting workers’ stress using deep learning 

Dear Dr. Zhang:

I'm pleased to inform you that your manuscript has been deemed suitable for publication in PLOS ONE. Congratulations! Your manuscript is now with our production department. 

Kind regards, 

on behalf of

Dr. Ashraf Imran 

Academic Editor

PLOS ONE